# Guardians of the Genome: BRCA2 and Its Partners

**DOI:** 10.3390/genes12081229

**Published:** 2021-08-10

**Authors:** Hang Phuong Le, Wolf-Dietrich Heyer, Jie Liu

**Affiliations:** 1Department of Microbiology and Molecular Genetics, University of California, Davis, CA 95616, USA; hphle@ucdavis.edu (H.P.L.); wdheyer@ucdavis.edu (W.-D.H.); 2Department of Molecular and Cellular Biology, University of California, Davis, CA 95616, USA

**Keywords:** BRCA2, homologous recombination, DNA repair, fork protection, genome stability

## Abstract

The tumor suppressor BRCA2 functions as a central caretaker of genome stability, and individuals who carry BRCA2 mutations are predisposed to breast, ovarian, and other cancers. Recent research advanced our mechanistic understanding of BRCA2 and its various interaction partners in DNA repair, DNA replication support, and DNA double-strand break repair pathway choice. In this review, we discuss the biochemical and structural properties of BRCA2 and examine how these fundamental properties contribute to DNA repair and replication fork stabilization in living cells. We highlight selected BRCA2 binding partners and discuss their role in BRCA2-mediated homologous recombination and fork protection. Improved mechanistic understanding of how BRCA2 functions in genome stability maintenance can enable experimental evidence-based evaluation of pathogenic BRCA2 mutations and BRCA2 pseudo-revertants to support targeted therapy.

## 1. Introduction

Genome instability is a fundamental cause of tumorigenesis, and the DNA damage response coordinates multiple DNA repair pathways at the frontline guarding genome integrity. BRCA2 is a prototypical caretaker gene, and its inactivation leads to genomic instability that is associated with oncogene activation and loss of tumor suppressor genes [1]. Thus, individuals who carry BRCA2 mutations have elevated risks for breast, ovarian, prostate, pancreatic, and other cancers [2,3]. BRCA2-deficient cells exhibit severe DNA repair phenotypes, including gross chromosomal rearrangements, accumulation of chromatid breaks, and hypersensitivity to genotoxins [4,5]. BRCA2 plays a conspicuous role as a mediator in homologous recombination (HR), a high-fidelity DNA repair pathway for double-strand breaks (DSB), and interstrand crosslinks (ICL) with additional functions in DNA replication fork support (Figure 1) [6,7]. In the last ten years, mounting evidence has demonstrated a direct role of BRCA2 in replication fork protection (RFP), both in an HR-dependent and HR-independent manner [8]. Recently, BRCA2 has also been shown to suppress other DSB repair pathways and prevent genomic rearrangements, independent of RAD51 [9]. BRCA2 engages in a multitude of protein interactions affecting its function as well as nuclear or chromosomal localization, including DSS1, PALB2, and EMSY (Figure 2) [10,11,12,13]. In this contribution, we will focus on the roles of BRCA2 in HR, replication fork protection, and DSB repair pathway control with a discussion of recent exciting discoveries concerning BRCA2-interacting proteins to extent earlier cogent reviews of BRCA2 and its roles in genomic stability maintenance [6,7,13,14].

## 2. BRCA2 as a Mediator of DNA DSB Repair

### 2.1. Role of BRCA2 in Homologous Recombination in Somatic Cells

Deficiency of BRCA2 leads to defective HR repair of chromosomal breaks [16,17]. A primary role of BRCA2 in genome stability maintenance is to facilitate the assembly of the nucleoprotein filament of RAD51, the central enzyme in HR. The RAD51 filament is the catalytic scaffold for homology search and DNA strand invasion, the central steps of HR (Figure 1). The 3′ ssDNA overhang generated by DSB resection sets the stage for the orchestrated turnover between initial occupancy of RPA, the eukaryotic ssDNA-binding protein, and later binding of RAD51 and additional proteins such as the RAD51 paralogs and RAD54 [14]. BRCA2 is required for this timely exchange to enable the formation of RAD51 filaments, as shown by biochemical reconstitution with purified full-length BRCA2 protein and supported by genetic and cell biological analyses [4,5,18,19,20,21,22]. BRCA2 is not known to play a direct role in the ensuing steps of D-loop formation, HR-associated DNA synthesis, and the processing of joint molecules to restore intact chromosomes (Figure 1). This section will review the current status of the structure of BRCA2 and discuss the recent progress in characterizing BRCA2 as a mediator protein focusing on three key biochemical functions: DNA binding, turnover of the RPA-ssDNA complex, and RAD51 loading.

#### 2.1.1. BRCA2 Structure

With 3418 amino acids (384 kDa), BRCA2 comes with considerable size organized in multiple functional domains and motifs. The N-terminal region contains two protein-interaction sites (for PALB2, EMSY), a DNA-binding site (termed N-terminal DNA-binding domain or NTD), and it is predicted to be intrinsically disordered (Figure 2A) [6,23,24,25]. The central domain of BRCA2 contains eight BRC repeats, spanning almost one-third of the protein, and constituting the primary interaction sites of RAD51 (Figure 2A) [6,26]. The complex C-terminal DNA-binding domain (CTD) is capable of binding both ssDNA and double-stranded DNA (dsDNA). The C-terminus also contains two nuclear localization signals (NLS) and an additional RAD51 interaction site (TR2) that specifically binds to RAD51 filaments (Figure 2A) [6,12,27,28,29]. Around these established domains, intrinsically disordered regions distribute across BRCA2 and include additional protein-interaction sites [23,24]. This arrangement allows for structural plasticity associated with conformational changes induced by BRCA2 binding partners to potentially regulate the functions of BRCA2 [23,24].

A high-resolution structure of the BRCA2 CTD in complex with ssDNA and DSS1, a small binding partner of BRCA2 (see below), has provided detailed information on how BRCA2 engages these two factors simultaneously with a helical domain and three oligonucleotide/oligosaccharide binding folds (OB folds) [12]. Interestingly, the ~800-residue mouse CTD displays an extended conformation reaching a length of 115 Å, with all the four domains packed sequentially and linearly (Figure 2B). A unique tower domain identified within the OB2 region features two long and antiparallel helices with a three-helix bundle on top. The structural similarity between the three-helix bundle and the helix-turn-helix dsDNA binding motif suggests that this region may support the non-specific dsDNA binding activity of BRCA2 [12]. DSS1 traverses through the helical domain to OB1 in two segments, while ssDNA tunnels through the base of the tower domain in OB2 to OB3. DSS1 and ssDNA locate on the opposite sides of this BRCA2 crystal structure making contact with different BRCA2 domains.

A low-resolution electron microscopic (EM) structure of full-length BRCA2 presents a kidney-bean shaped dimer (Figure 2C), and it could bind two oppositely arranged clusters of RAD51 on top [30]. The symmetrical dimer in antiparallel orientation would indicate that the two CTDs bind ssDNA equally to initiate two RAD51 filaments in opposite directions [30]. However, direct visualization of the BRCA2-RAD51-ssDNA filaments indicates a 3′ to 5′ growth of RAD51 with BRCA2 at the 3′ terminus, which was interpreted that only one set of bound RAD51 proteins would be productive to form filaments on DNA [30]. In living cells and in the absence of genotoxic stress, GFP-tagged BRCA2 exists as a mixture of monomers, dimers, and multimers, based on single-particle tracking and fluorescence correlation spectroscopy [31]. Scanning force microscopy revealed large oligomers of BRCA2 in branch-like and elongated structures with significant heterogeneity [32]. The structural plasticity of BRCA2 is further evidenced by low temperature-induced conformational changes towards more globular structures [32]. Two types of BRCA2 self-interactions have been identified to be responsible for the formation of dimers and large oligomers, a strong N-to-C terminal interaction in a head-to-tail mode and a weaker N-to-N terminal interaction in a head-to-head mode [23]. DSS1 and ssDNA antagonize this BRCA2 self-association and preferentially stabilize in an independent fashion the monomeric form of BRCA2, as visualized by a low-resolution EM structure of the monomeric BRCA2-DSS1-ssDNA complex (Figure 2C) [23]. Recent scanning force microscopy studies show that also binding to RAD51 dramatically decreases the population of large BRCA2 oligomers and increases the proportion of monomeric BRCA2 from less than 10% to almost 75% for the full-length protein [24]. These results strongly suggest that binding partners and the DNA substrate modulate the conformation of BRCA2 to regulate its functions. Many questions about the structure of BRCA2 remain and more high-resolution studies are needed to understand how individual domains organize and interact in the context of the full-length BRCA2 protein.

#### 2.1.2. DNA Binding Activities of BRCA2

Human BRCA2 contains two distinct DNA-binding regions, but their individual specificity and potential cooperation are not fully understood [12,25,50,51]. The amino-terminal DNA-binding domain (NTD) (residue 267–350) consists of a zinc finger-poly(adenosine diphosphate-ribose) polymerase (PARP)-like domain and binds to different DNA structures (ssDNA, dsDNA, tailed DNA, and gapped DNA) with similar affinity [25]. The CTD (residue 2474–3190) is significantly more complex and binds these DNA substrates with different affinities [12,51]. The OB-folds bind ssDNA and the apical 3 bundle helix in the tower domain was proposed to bind dsDNA. A protein fragment encompassing the entire CTD to the end of the protein (residue 2153–3418) displays a much higher binding affinity towards 3′-tailed DNA than an NTD fragment (residue 1–992) [50]. Additions of one, four, or eight BRC repeats do not change the affinity significantly [50]. In contrast, a smaller CTD fragment (residue 2474–3190) shows a significantly lower binding affinity in direct comparison to a different NTD fragment (residue 250–500) [25]. Interestingly, both reports use 3′-tailed DNA substrates with identical lengths, eliminating this as a potential reason for the apparent difference. A third CTD fragment (residue 2477–3194), fused with two BRC repeats (BRC3/4), shows a strong binding preference towards ssDNA compared to dsDNA [51]. This BRC3/4-CTD construct show saturated binding of ssDNA at a protein concentration of 500 nM [51], while an almost identical CTD construct without the BRC repeats shows very little binding of ssDNA at the same protein concentration of 500 nM [25]. It’s not clear whether these conflicting results are a result of different in vitro conditions or a reflection of different properties of the constructs. However, both constructs show a consistent binding preference towards ssDNA compared to dsDNA substrates [25,51], analogous to the DNA binding properties of full-length BRCA2 [18]. The DNA binding-site sizes of the individual DNA binding regions (NTD, CTD with its individual DNA binding modules) and the full-length BRCA2 remain to be determined. Full-length BRCA2 and a mouse CTD fragment show increasing affinity towards ssDNA with an increasing length of the DNA substrate but with a different minimum length requirement [12,30]. Longer ssDNA oligonucleotides were also more effective in disrupting the BRCA2 N-N and N-C terminal self-associations [23]. This may suggest that the CTD can engage various numbers of the OB folds when binding to ssDNA. While BRCA2 has been shown to bind to ssDNA and dsDNA, its complexity and size have complicated the analysis of the basic DNA binding properties of this important protein. More studies are required to understand the individual contributions by and potential coordination between the NTD, CTD, and the CTD subregions (OB-folds, tower domain) in the DNA-binding activity of the full-length BRCA2 protein.

The mechanism of recombination mediator proteins has been highly conserved from bacteria to eukaryotes [52]; *E. coli* RecFOR and the *U. maydis BRCA2* homolog (Brh2) serve as paradigms to understand the molecular actions of BRCA2 [53,54,55]. RecFOR and Brh2 recognize the ss/dsDNA junctions and load RecA and RAD51, respectively, onto the junctions to start one-directional filament growth from 5′ to 3′ [54,55,56]. Yeast Rad52 forms a multimeric ring-like structure and wraps ssDNA to destabilize RPA-ssDNA interaction for RAD51 loading [57,58,59]. The tower domain identified in the crystal structure of the BRCA2 CTD indicates a potential binding activity towards dsDNA [12], suggesting that BRCA2 could initiate a unidirectional RAD51 filament growth from an ss/dsDNA junction [12,53,55]. Unexpectedly, full-length BRCA2 displays no preference towards ss/dsDNA junctions and binds with almost equal affinity to 3′-tailed, 5′-tailed, and ssDNA, although tailed DNAs are the preferable substrates in RAD51-mediated strand exchange [18]. It is unknown whether some protein partners might endow structure-selective DNA-binding to BRCA2 towards ss/dsDNA junctions or other more complex DNA intermediates, potentially through altered DNA engagement of different DNA-binding domains (DBD). Overall, it is likely that the DNA-binding activities of BRCA2 and the potential for modulation by its protein partners might be uniquely fitted to its versatile activities in HR and fork stabilization.

#### 2.1.3. Mechanism of Nucleation of RAD51 Filaments on RPA-Coated ssDNA

The high affinity and strong cooperativity of RPA towards ssDNA allow rapid binding of the newly exposed 3′ overhang after DSB resection and inhibits binding of RAD51 [60]. The logic of this sequence is that RPA melts secondary structure to present RAD51 with a fully single-stranded DNA substrate. As RAD51 has only a slightly higher affinity for ssDNA than dsDNA [61,62], this sequence avoids the formation of RAD51:dsDNA filaments that are inactive in homology search and DNA strand invasion [37,63,64]. Mechanistically, BRCA2 facilitates RAD51 filament formation on RPA-coated ssDNA [18,19]. The BRC repeats and the CTD are required for this function [51], but the exact mechanism remains to be determined. The yeast Rad52 provides a paradigm for mediator function and RPA displacement [65,66,67]. Rad52 directly interacts with RPA in a species-specific fashion; it stimulates RAD51-dependent DNA strand exchange activity when ssDNA is pre-coated with RPA but not its bacterial homolog, *E. coli* SSB [65,66,67]. These results imply that the specific protein-protein interaction between Rad52 and RPA is required for the RPA-RAD51 exchange on ssDNA. Additionally, crystal and EM structures of the RAD52 protein reveal a ring-like oligomeric structure that wraps ssDNA to weaken the RPA-ssDNA interaction through kinking the DNA and thus promoting RAD51 filament formation [58,59,68]. Single-molecule imaging studies show that Rad52 binds tightly to the RPA-ssDNA complex, and small clusters of Rad52 and RPA could persist when RAD51 forms filaments on ssDNA [69]. Overall, the initial weakening of the RPA-ssDNA complex represents the action of the mediator protein such as yeast Rad52. Its binding to RAD51 increases the local RAD51 concentration to support filament nucleation. The following large-scale displacement of RPA is the result of cooperative RAD51 filament growth, although some RPA clusters might remain on the resected ssDNA substrate. Consequently, yeast Rad52 is central for HR which requires its protein-protein interaction with RPA [57,70,71,72].

Human RAD52 shows no mediator activity in standard reconstituted biochemical reactions [18], except at a sub-stoichiometric RAD51 concentration [73]. Deletion or depletion of RAD52 in mammalian cells causes no major HR defect, and it is generally accepted that BRCA2 is the major mediator in human cells [53,74]. However, unlike yeast Rad52, BRCA2 has no direct interaction with RPA but still stimulates RAD51 filament formation and DNA strand exchange activity in the presence of RPA inhibition [18,19]. More studies of BRCA2 are needed to determine how BRCA2 enables RAD51 filament formation without direct protein interaction with RPA.

#### 2.1.4. BRCA2 Mediates RAD51 Loading

The eight BRC repeats serve as the primary interaction motifs for RAD51 and are highly conserved among mammalian species [75,76]. Interestingly, BRCA2 homologs from organisms such as *C. elegans* (BRC-2) or *U. maydis* (Brh2) contain only a single BRC repeat. Cellular studies provide support for the physiological importance of the interactions between BRC repeats and RAD51 in the HR repair. The eight BRC repeats are encoded by exon 11, and BRCA2 mutant mice with a disrupted exon 11 show embryonic lethality and γ-radiation hypersensitivity [4]. These BRCA2-deficient *Brca^Brdm1/Brdm1^* mice embryos, lacking amino acids 626–1437 of exon 11 in BRCA2, display strong developmental arrest after 6.5 days of gestation, and no homozygous mutant cell lines could be generated, supporting the key roles of BRCA2 and its interaction with RAD51 [4]. The second line of evidence is from the dominant-negative effects of BRC repeats in cell lines expressing wild-type BRCA2. Exogenous expression of a single BRC repeat, BRC1, BRC3, or BRC4, disrupts the BRCA2-RAD51 interaction and reduces IR-induced RAD51 focus formation leading to HR-deficiency [17,21,77,78]. In summary, the interaction between BRC repeats and RAD51 is key for BRCA2 to localize RAD51 to the damaged site.

Biochemical studies with individual BRC repeats demonstrate that BRC1-4 binds free RAD51 with high affinity, stimulates the formation of RAD51-ssDNA complexes, and prevents RAD51 binding onto dsDNA, while BRC5-8 stabilize RAD51-ssDNA nucleoprotein filaments [37,38]. An FxxA motif, identified from the high-resolution crystal structure of the BRC4-RAD51 complex, binds RAD51 by molecular mimicry of the oligomerization interface between individual RAD51 monomers [26]. The TR2 region was mapped to the extreme C-terminus as a secondary RAD51 binding site (Figure 2A). TR2 binds RAD51 filaments and regulates the function of BRCA2 in fork stabilization rather than HR through phosphorylation of the S3291 residue (Figure 2A) [27,28,29,79]. It remains unknown whether individual BRC repeats and the TR2 region cooperate in RAD51 recruitment, considering that the binding affinity of full-length BRCA2 to RAD51 is about 100-fold higher than that of an isolated BRC repeat [18]. Surprisingly, BRC5-8-DBD, a protein fragment containing BRC5-8 and the entire CTD (residues 1596–3418), shows enhanced stimulation in RAD51-catalyzed DNA strand exchange and almost full complementation of HR repair deficiency and RAD51 IR-induced foci formation in human BRCA2 knockout cell lines, compared to the same DBD constructs fused with a single BRC4, BRC repeats 1–4, or BRC repeats 1–8 [50]. It remains unclear whether BRC repeats could potentially antagonize each other, and further studies are needed to address these issues.

### 2.2. Role of BRCA2 in Meiotic Homologous Recombination

Besides maintaining genome integrity in somatic cells, HR is essential for meiotic chromosome segregation by generating interhomolog crossovers and achieving genetic variation among the offspring [80]. Thus, defective HR repair in meiosis can cause infertility, miscarriage, and chromosomal missegregation leading to genetic diseases [81]. The meiotic HR process resembles the mitotic one with some major adaptations. First, DSBs are initiated as programmed events at hot spots by the conserved SPO11 transesterase [82]. Second, DMC1, a meiosis-specific RAD51 paralog, catalyzes meiotic recombination and RAD51 plays an accessory role to DMC1 [83,84,85]. DMC1 gains the ability to generate heteroduplex duplex DNA joints with mismatch-containing base triplets, whereas RAD51 cannot [86]. This specificity of the DMC1 lineage may explain why DMC1 catalyzes meiotic HR instead of RAD51. Third, the preferred repair template is switched from sister chromatids to homologous chromosomes [87]. Forth, crossovers are the designated outcome for meiotic HR, a process that is under tight regulation. The number of crossovers is limited to the range of one per chromosome to one per chromosome arm, to ensure accurate segregation of homolog chromosomes and genetic diversity [81,88]. Lastly, a number of meiosis-specific factors are added to the core HR machinery to provide an extra layer of regulation of meiotic recombination, as discussed in Part V (Figure 1) [81].

Emerging evidence assigns BRCA2 a key role also in meiotic HR. BRCA2 exhibits high expression levels in mouse and human testis [89,90]. It localizes to the unsynapsed axial element and unpaired X and Y chromosomes in late zygonema–early pachynema in human spermatocyte nuclei [91]. Loss of BRCA2 function leads to infertility in all organisms studied. BRCA2-deficient mouse spermatocytes show dramatically reduced RAD51 and DMC1 foci at meiotic chromosomal axes, failing in gametogenesis at the early prophase I stage [92]. However, only enrichment of RAD51, but not DMC1, was detected in immunoprecipitates of BRCA2 from mouse testis extracts, suggesting a weak or transient interaction between BRCA2 and DMC1 [93]. DMC1 is a germline-specific gene and shares significant similarities to RAD51 in biochemical activity and protein structure [81,83,84,85]. Direct physical interaction between BRCA2 and DMC1 was reported for plants and humans [18,39,94,95,96]. In *Arabidopsis*, yeast two-hybrid assays show that the interactions of BRCA2 with RAD51 and DMC1 occur via different BRC repeat regions, *Brc2* for DMC1 and Brc4 for RAD51 [94,95]. In humans, the initial DMC1 interaction site was mapped to residues 2386–2411 [96], but mutation of a conserved key residue (Phe2406) had no effect on meiotic progression and gametogenesis in mice [97]. Later, it was shown that all individually purified BRC repeats could bind to DMC1 with varying affinity and stimulate its activity in DNA joint molecule formation to different extents [39]. Consistent with this observation that BRC repeats interact with RAD51 and DMC1, purified full-length BRCA2 stimulates RAD51- and DMC1-catalyzed DNA strand exchange activities in a remarkably similar fashion [39]. Additionally, DMC1 also shows interaction with the TR2 domain at the C-terminus of BRCA2 [96]. However, transgenic mice carrying a C-terminal BRCA2 TR2 deletion are fertile with no meiotic phenotypes [98,99], suggesting that this RAD51 interaction domain is not essential for meiosis. In summary, genetic and cellular studies support a mediator function of BRCA2 to facilitate RAD51 and DMC1 recruitment required during meiotic recombination of programmed DSBs. However, it remains unknown whether one BRCA2 molecule could interact with RAD51 and DMC1 simultaneously to initiate a short hetero-filament on ssDNA as proposed in [39] and how the additional interactions at TR2 would impact the organization of the mixed filaments of RAD51 and DMC1.

## 3. Role of BRCA2 in Replication Fork Restart

Stalled replication forks need to be restarted or converge with another fork to complete genome duplication before cells enter mitosis to prevent catastrophic DNA breakages and eventually cell death. Endogenous DNA lesions can trigger fork blockage or helicase-polymerase uncoupling and thus fork stalling, accompanied by the accumulation of ssDNA gaps [100,101]. Such DNA lesions include base damages, intra- and inter-strand crosslinks (ICLs), and protein-DNA complexes. Fork stalling lesions can also be induced by exogenous sources, such as UV radiation, aphidicolin, and MMC, or replication inhibitors such as hydroxyurea [100,101]. Cells utilize several different mechanisms to restart stalled forks, including fork reversal, fork breakage, translesion synthesis (TLS), and template switching [100,102]. This section will discuss the HR-dependent and HR-independent functions of BRCA2 in supporting replication fork restart, with a focus on fork reversal.

### 3.1. HR-Independent Role of BRCA2 in Replication Fork Protection

The function of BRCA2 in replication fork support had been linked to its stimulation of RAD51-mediated DNA strand invasion, a key reaction in the HR pathway to repair toxic intermediates produced when replication forks are stalled or broken, such as replication gaps and one-sided DSBs [100]. Surprisingly, recent data demonstrated a novel and HR-independent role of BRCA2 in stabilizing replication forks and protecting them against nuclease degradation [79]. This section will discuss recent progress focusing on the fork protection role of BRCA2 (Figure 3).

Fork reversal refers to the active remodeling of stalled replication forks after leading strand blockage, during which a so-called chicken foot structure can be formed by annealing the two newly synthesized sister strands leading to a DSB end (Figure 3). This unique fork architecture was visualized by electron microscopy, occurring at an astounding frequency of 15–30% with all tested genotoxic treatments [101]. The high frequency of fork reversal substantiates its status as a likely physiological response to replication stress. There are two distinct pathways to remodel the forks actively and both require RAD51 but involve different motor proteins [103]. One route involves SMARCAL1 [104,105,106], ZRANB3 [104,107], and HLTF [108,109], all ATPase-dependent DNA translocases of the SWI2/SNF2 family, while the other involves the F-box DNA helicase FBH1, a UvrD homolog [110]. Reversed forks are targeted by multiple nucleases, including MRE11, EXO1, and DNA2 [79,111,112,113,114,115], and require fork protection to prevent nucleolytic degradation. As shown in Figure 3, reversed forks from these two remodeling pathways are attacked by different nucleases and recruit distinct replication fork protection (RFP) factors [103]. BRCA2 is essential to protect the fork against MRE11-dependent degradation, together with FANCD2 and ABRO1 [103] (Figure 3A1). In BRCA2-deficient cells, the replication fork undergoes massive degradation by MRE11 after HU treatment, as evidenced by DNA fiber assays [79,112,115], electron microscopic imaging [111], and 2D-gel electrophoresis [116]. Fork degradation could be suppressed by either BRCA2 back-complementation or MRE11 inhibition [79]. Depletion of SMARCAL1, ZRANB3, and HLTF, but not FBH1, restores fork stability in BRCA2-deficient cells, suggesting that they participate in two different subpathways (Figure 3A) [103,112,117]. The recruitment of BRCA2 to stalled replication forks is dependent on BRCA1 and PALB2, as depletion of either one leads to fork destabilization [117,118,119]. Additionally, BRCA2 protects the fork from degradation by EXO1, but not DNA2, since knockdown of EXO1, but not DNA2, restores the fork stability in BRCA2-deficient cells [111]. Instead, DNA2-mediated degradation is suppressed by 53BP1, FANCA, BOD1L, and VHL in the second pathway [103] (Figure 3A2). It is unclear what differentiates the two chicken foot pathways and whether there are alternate chicken foot structures.

The independence of the roles of BRCA2 in fork protection and HR repair is supported by the separation-of-function mutant BRCA2 S3291A. This mutant retains full proficiency in HR repair [79] but fails to rescue stalled fork stability in BRCA2-deficient cells [79,112]. Mutations of S3291 to alanine or glutamic acid or phosphorylation at this site by CDKs disrupts the RAD51 interaction of TR2 [27,43] and promotes RAD51 filament disassembly [120], indicating that the fork protection role of BRCA2 is regulated by CDKs and require interaction with RAD51. Indeed, in BRCA2-proficient cells, overexpression of BRC4 disrupts RAD51 filaments [78] and promotes fork degradation [79], whereas, in BRCA2-deficient cells, overexpression of the RAD51 K133R mutant leads to stable RAD51 filaments [121] and rescue of RFP [79]. These data suggest the essential activity of BRCA2 in RFP and HR remains identical, recruiting and stabilizing RAD51 filaments. The initial recruitment of RAD51 to the nascent DNA strand is essential for fork reversal since depletion of RAD51 antagonizes fork degradation by avoiding the formation of reversed fork [114]. RAD54 deficiency does not affect the stability of stalled replication forks [79], further supporting that fork reversal is independent of the HR pathway.

In addition to preventing resection of the stalled forks, emerging data suggest that BRCA2 also contributes to fork recovery in a RAD51-independent manner. Together with PALB2, BRCA2 stimulates the recruitment of Polη to stalled and collapsed replication forks through direct protein interactions. In vitro, both purified BRCA2 and PALB2 interact with Polη and stimulate its DNA synthesis on a short oligonucleotide construct mimicking D-loops [122]. However, BRCA2-mediated fork restart might represent a small proportion (around 5%) in human cells [79], and the majority of the events are likely through the RAD51 [123] or MUS81/POLD3 pathways [111] (Figure 3).

### 3.2. HR-Dependent Role of BRCA2 in Template Switching

Template switching relies on HR to bypass DNA lesions utilizing the newly synthesized daughter strand on the sister chromatid as a template for replication fork restart to repair postreplicative gaps [124] (Figure 3). Moreover, extended treatment with hydroxyurea leads to prolonged blocking and breaking of replication forks, which requires the RAD51-mediated HR pathway to restart from one-sided DSBs [123]. MUS81 is responsible for generating these one-sided DSBs, and BRCA2 is expected to function as a mediator protein to facilitate RAD51 filament formation to catalyze homology search and DNA strand invasion (Figure 3B) [123,125,126]. An alternative pathway of stalled fork processing is fork repriming, which mainly relies on TLS polymerases to replicate through a damaged template (Figure 3C) or by skipping the lesion to reinitiate replisome downstream of the lesion by PrimPol in humans (Figure 3D) [100,102].

## 4. DSB Repair Pathway Control

To repair a single DNA DSB, HR competes with three other pathways in the cell: classic non-homologous end-joining (NHEJ), microhomology-mediated end-joining (MMEJ), and single-strand annealing (SSA) [127]. Recent evidence indicates an additional role of BRCA2 in DSB repair pathway choice, independent of RAD51. Inactivation of BRCA2, but not RAD51, accelerates the dissolution of IR-induced RPA foci and causes gross chromosome instability as well as massive nuclear fragmentation [9]. In BRCA2-deficient cells, the majority of IR-induced DSBs are repaired by either NHEJ or POLθ-mediated MMEJ, as inactivation of both pathways almost fully suppresses the nuclear fragmentation phenotype [9]. In the context of these BRCA2-depleted cells, SSA plays a negligible role and is not required to stabilize RPA foci and suppress nuclear fragmentation [9], even though its frequency is substantially elevated when measured by reporter assays [9,128,129]. It was proposed that BRCA2 is rapidly recruited to the damaged sites and block the access of NHEJ and MMEJ factors [9]. The synthetic lethality between loss of BRCA2 with the loss of either RAD52 (SSA) or POLθ (MMEJ) supports the competitive nature of these pathways in DSB repair [130,131]. However, more studies are needed to uncover the exact mechanism underlying the synthetic lethality between BRCA2, RAD52, and POLθ to establish the underpinning for new therapeutic strategies targeting RAD52 or POLθ in BRCA2-deficient tumors [132].

Unique patterns of mutations can be identified in cancer genomes, and such mutational signatures are caused by defective DNA replication and repair functions [133]. Tumor samples with defective BRCA2 display the defined HR-defective mutational signature 3 with single-base substitutions (SBSs), ID6 with small insertions and deletions, and structural variants with non-clustered deletions < 100 kb and reciprocal translocations [133,134,135,136,137]. As predicted, impaired HR repair generates a seemingly random and equal distribution of SBSs across the genome. The increase in microhomology-mediated deletions (5–50 bp) indicates that NHEJ and MMEJ dominate the DSB repair process in the absence of HR [133,134,135,136]. These mutational events are likely the consequence of the POLθ and RAD52-mediated repair pathways [9,138]. Structural variants with medium-length deletions are proposed to be the consequence of BRCA2 defects in fork protection and restart [137]. In conclusion, a mechanistic understanding of the role of BRCA2 in DSB repair pathway choice is critical to developing novel therapeutic targets based on synthetic lethal interactions with BRCA2 deficiency to enhance treatment efficacy.

## 5. Regulation of BRCA2 Function via Its Binding Partners

Multiple interaction partners of BRCA2 have been reported with potential functions including DNA repair, fork stability, and cell-cycle control (Table 1). Multiple post-translational modifiers interact with BRCA2, including CDK2 (Cyclin-dependent kinase), PLK1 (Polo-like kinase), BRCA1 (E3 ubiquitin ligase), USP21 (deubiquitinase), and BRAF35 (histone deacetylase) (Figure 2) [33,43,44,139,140,141,142,143]. In addition, BRCA2 was proposed to serve as a scaffold to organize the interaction between p/CAF acetyltransferase and BubR1 kinase [144]. Moreover, BRCA2 has been proposed to play an HR-independent role in cytokinesis as a scaffold protein to form different complexes with Filamin A, CEP55, and NM-IIC in the midbody to facilitates proper abscission [145,146]. In this section, we limit the discussion to a few selected BRCA2-binding partners with confirmed interaction sites and demonstrated involvement in HR repair and RFP.

### 5.1. DSS1, a Versatile Structural Modulator

DSS1 is a unique partner of BRCA2 with only 70 amino acids, containing two acidic stretches comprised mostly of aspartate and glutamate. The biological functions of DSS1 are versatile, and *DDS1* features an expansive interactome, not limited to proteins in HR repair [166]. As a structural component, DSS1 connects Sac3 and Thp1 in the TREX-2 transcription export complex, Csn12 and Thp3 in a transcriptional regulatory complex, and Rpn3 and Rpn7 in the 26S proteasome [152,153,154,155,156,157]. Sem1, the yeast DSS1 homolog, sequesters Rpn3 and Rpn7 into a complex through its two acidic stretches as a molecular tether [158].

DSS1 was found to be required for the crystallization studies of the BRCA2 CTD and binds across OB1 and OB2 (Figure 2) [12]. Mutational studies support the physiological importance of the BRCA2-DSS1 interaction since loss of DSS1 phenocopies a BRCA2 defect in mammals and fungi [159,160]. In mammals, depletion of DSS1 dramatically decreases DNA damage-induced RAD51 focus formation and HR repair efficiency while increasing chromosome aberrations [159,161]. In fungi, mutants deficient in DSS1 show hypersensitivity to IR and decreased HR repair efficiency, identical to mutants of RAD51 and brh2 (fungal BRCA2 homolog) [160]. DSS1 contributes to the stability of BRCA2 in cells [162,163]. Defective binding of DSS1 leads to reduced HR activity of BRCA2 mutant constructs (K2630A and K2630D) containing the CTD and one to four BRC repeats [49]. A point mutation at another site on BRCA2, D2723H, is cancer-prone and leads to cytoplasmic mis-localization of BRCA2 due to wrongly exposed nuclear export signals as a consequence of defective DSS1 binding [48]. In vitro, the addition of purified DSS1 results in stimulation of full-length BRCA2 in RAD51 filament formation on RPA-covered ssDNA [19]. Biochemical studies have led to multiple models of how DSS1 regulates BRCA2 activity, including protein stability, DNA binding, monomer-dimer transition, and the BRCA2-RAD51 interaction [159,162,163,188,189,190,191].

Recently, DSS1 has been proposed to act as a DNA mimic to bind RPA and weaken the RPA-ssDNA interaction directly [164]. This mechanism represents an additional, BRCA2-independent role of DSS1 in HR repair since DSS1 stimulates RAD51-ssDNA filament formation and RAD51-catalyzed DNA strand exchange activity in the presence of RPA inhibition in the absence of BRCA2 [18,19]. DSS1 also binds RAD52 directly and changes its oligomeric conformation. As a result, DSS1 modulates the DNA-binding activity of RAD52 to stimulate its ssDNA single-strand annealing (SSA) activity, reflecting their participation in the SSA and break-induced replication (BIR) pathways [165].

The role of DSS1 in controlling the BRCA2 oligomer to monomer transition has been discussed above under Section 2.1.1. *BRCA2 structure*, but the exact mechanism still needs to be established. In sum, DSS1 has become a critical factor in genome stability acting at least at three different levels, in conjunction with BRCA2, as a DNA mimic, and modulating RAD52. This multitude of functions and the involvement of DSS1 in processes unrelated to genome stability complicates the interpretation of genetic data and requires further mechanistic studies.

### 5.2. The Partner and Localizer of BRCA2: PALB2

PALB2 (Partner and Localizer of BRCA2) was discovered as a BRCA2-interacting protein and later characterized as a tumor suppressor and a member (FANCN) of the Fanconi Anemia (FA) pathway, which also includes BRCA2 (FANCD1) and other HR proteins (BRCA1-FANCS, RAD51-FANCR, RAD51C-FANCO, BRIP1-FANCJ, XRCC2-FANCU) (Figure 2A) [11,192,193,194,195,196]. FA patient-derived fibroblasts with PALB2 deficiency show hypersensitivity to cross-linking agents, gross depletion of chromatin-bound BRCA2, and loss of MMC-induced RAD51 foci [195]. The colocalization of IR-induced PALB2 foci with BRCA1, BRCA2, and the DNA damage marker γH2AX suggested a linker role of PALB2 to recruit BRCA2 to BRCA1-bound DSB sites [11]. PALB2 interacts with BRCA1 through its coiled-coil domain at the N-terminus and with BRCA2 through the WD40 domain at its C-terminus [11,141,142,148]. The PALB2 binding domain has been mapped to the residues 21–39 of BRCA2. A high-resolution crystal structure of this BRCA2 peptide in complex with the C-terminal fragment of PALB2 identified the key interaction residues (Figure 2A) [11,34]. The N-terminal coiled-coil domain is also the self-association domain of PALB2, and a transition from PALB2 oligomers to the monomeric BRCA1-PALB2-BRCA2 complex was proposed to occur upon DNA damage-induced and ATM/ATR-dependent phosphorylation of PALB2 [197,198,199,200,201]. In S/G2 cells, PALB2 may also be recruited to DSBs through direct interaction with the RING finger (RNF) E3 ubiquitin ligase RNF168 in a BRCA1-independent manner [202,203,204]. Purified PALB2 binds directly RAD51 in vitro and stimulates RAD51-catalyzed D-loop formation and DNA strand exchange activities in the absence of BRCA2 [205,206]. This stimulation by PALB2 could be further enhanced by a truncated version of BRCA2 [206]. It was proposed that the direct stimulation of RAD51 by PALB2 might be an additional function and occur only after the successful formation of a BRCA1-PALB2-BRCA2 complex [192]. Indeed, a BRCA2 point mutation (W31C) that abolishes the interaction with PALB2 has dramatically decreased HR repair efficiency [34,49]. Similarly, PALB2 is vital for the RFP activity of BRCA2. Upon RPA phosphorylation, PALB2 binds and recruits BRCA2 to RPA-covered ssDNA to initiate the recovery of replication forks in cells experiencing replication stress [119]. In summary, PALB2 is a recruiting factor of BRCA2 to the damaged DNA sites and the stalled replication forks, whereas it remains largely unknown whether PALB2 affects the DNA- and protein-binding activities and specificities of BRCA2.

### 5.3. EMSY, a Potential Negative Regulator of BRCA2

EMSY was discovered as a BRCA2-interacting protein through yeast two-hybrid screens with small overlapping fragments of the N-terminus of BRCA2. The EMSY binding site has been mapped to the residues 23–44 of BRCA2 which overlaps with the PALB2 binding site (residues 21–39) (Figure 2A) [10,34]. Initially, EMSY was proposed to be a transcriptional regulator of BRCA2 through binding to a BRCA2 region that could function as a transactivation domain in the yeast two-hybrid system [10,207]. Some evidence supported the role of EMSY as a transcription factor. The ENT domain of EMSY shares structural similarity to the DNA-binding domain of homeodomain proteins [150]. EMSY interacts with chromatin-regulating proteins such as HP1 [10,149] and suppresses the expression of miR-31, an antimetastatic microRNA [151]. However, the role of BRCA2 in transcriptional regulation has not been confirmed, and instead, EMSY may have a direct function in the DNA damage response. The EMSY gene is frequently amplified in ovarian, breast, and other cancer types, and its amplification is associated with a poor prognosis and adverse outcomes in breast cancer [10]. EMSY amplification in sporadic breast and ovarian cancers resembles BRCA2 mutations in hereditary cancers in terms of the spectrum of pathology [10]. The effect of EMSY amplification in cancers could reflect a direct role in HR via BRCA2 interaction. Indeed, EMSY colocalizes with damage-induced γH2AX foci [10]. Overexpression of the N-terminal fragment of EMSY (1–478) inhibits spontaneous and DSB-induced HR, potentially by inhibiting the BRCA2-RAD51 or BRCA2-PALB2 interactions [208]. Consistently, overexpression of full-length EMSY suppresses HR and damage-induced RAD51 foci formation which requires the T207 phosphorylation site on EMSY [209]. These data suggest that amplification or overexpression of EMSY has a similar effect to BRCA2-inactivating mutations. One possibility is that the overlapping binding sites of PALB2 and EMSY on BRCA2 lead to antagonism. EMSY may also play a BRCA2-independent role in HR. Overexpression of the C-terminal fragment of EMSY, which does not bind to BRCA2, reduces HR efficiency to a greater level than overexpression of the N-terminal fragment or full-length EMSY [209]. The interference with HR could be a result of transcriptional regulation of the RAD51 gene since overexpression of full-length EMSY reduces RAD51 mRNA levels [209].

However, data from cancer patients and cancer cell lines are not consistent with the cellular data based on exogenously overexpressed EMSY in terms of the role of EMSY in HR. First, endogenous EMSY amplification in cancer cell lines does not correlate with EMSY mRNA and protein overexpression and show no impairment of HR and no sensitivity to PARPi and cisplatin treatment [210]. Second, an EMSY-amplified ovarian cancer cell line has normal RAD51 focus formation in response to DNA damage and RAD51 focus formation was not affected by knockdown of EMSY [209]. Conflicting results have been reported that four ovarian cancer cell lines with EMSY amplification are sensitive to PARPi [211]. Overall, more studies need to address the conflicting reports and evaluate whether EMSY is a *bona fide* inhibitor of HR repair.

### 5.4. MEILB2/HSF2BP-BRME1: A Meiotic Partner with Relevance for Cancer

MEILB2/HSF2BP (Meiotic localizer of BRCA2; HSF2 binding protein) was first discovered as the binding partner of heat shock factor 2 (HSF2) by yeast two-hybrid screening with specific expression in testis [212]. Later, it was defined as a meiotic prophase I-specific chromosomal axis-associating protein and identified as a novel meiosis-specific BRCA2-binding protein in mice [40,41]. MEILB2 binds to BRCA2 directly via a short 223 amino acid region (residues 2117–2339) located between the last BRC repeat and the helical domain (Figure 2) [40]. An independent study mapped the binding site to residues 2270–2337 and showed that a purified BRCA2 fragment (Gly2270-Ala2351) could stably interact with purified untagged hHSF2BP in vitro, further corroborating the direct interaction between BRCA2 and MEILB2 [41]. Loss of MEILB2 abolishes RAD51 and DMC1 focus formation in spermatocytes resulting in male sterility [40]. Five groups reported another meiosis-specific protein in complex with BRCA2 and MEILB2, named BRME1 (BRCA2 and MEILB2-associating protein 1) in mice and humans [93,213,214,215,216]. The C-terminus of BRME1 binds to the α-helical N-terminus of MEILB2, stabilizing MEILB2 by preventing its self-association [93]. The C-terminus of MEILB2 interacts with BRCA2, enabling MEILB2 to serve as a platform to form the BRCA2-MEILB2-BRME1 ternary complex in vitro and in vivo [93,213]. The meiosis-specific ssDNA binding complex SPATA22-MEIOB facilitates, but is not required, to recruit MEILB2-BRME1 and thus BRCA2 to ssDNA, since both MEILB2 and BRME1 foci were still detectable but significantly weaker in *Spata22^−/−^* and *Meiob^−/−^* spermatocytes [93]. The BRCA2-MEILB2-BRME1 complex is required for the assembly of RAD51 and DMC1 at meiotic DSBs during spermatogenesis since deletion of any component in the complex diminishes RAD51 and DMC1 foci formation at recombination sites, causing impaired DSB repair and infertility [40,92,93,213]. Additionally, the sexual dimorphic phenotype in both *Meilb2* and *Brme1* knockout mice is limited to males, indicating an additional mechanism for BRCA2 recruitment in meiosis in females [40,213,215]. MEILB2 and BRME1 are highly expressed in many human cancer tissues with a similar expression pattern [41,93]. Ectopic expression of MEILB2 and BRME1 in somatic cells largely abolish MMC-induced RAD51 foci formation, and this inhibition of ICL repair depends on the ability of MEILB2 to bind BRCA2 [93,167]. Overall, the MEILB2-BRME1 complex facilitates the recruitment of BRCA2 in meiotic HR, while it inhibits BRCA2-mediated HR in mitosis. MEILB2 may represent a meiotic counterpart of PALB2, but PALB2 itself is required for meiosis in sex chromosome synapsis and fertility [217]. It remains an open question how MEILB2 and PALB2 could participate in meiotic HR together and what could be the functional distinction between these two proteins in meiosis.

### 5.5. SYCP3, a Cancer Testis Antigen with a Structural Role in Synaptonemal Complex

SYCP3 is a component of the synaptonemal complex, an eminent protein structure along the entire length of maternal and parental chromosomes to promote meiotic crossover formation and accurate chromosome segregation [168]. In meiosis, BRCA2, RAD51, and DMC1 show punctate foci on SYCP3-marked axial elements [91,92,93]. Normal expression of SYCP3 is limited to meiotic cells, but emerging evidence shows an abnormal expression of SYCP3 in many cancers [169]. Like MEILB2-BRME1, the ectopic expression of SYCP3 in somatic cells has a negative effect on HR repair. Expression of SYCP3 in mitotic cells leads to accumulation of spontaneous DSBs, decreased formation of IR-induced RAD51 foci, hypersensitivity to DNA damages, and increased aneuploidy [170]. Coimmunoprecipitation studies show that mitotically expressed SYCP3 interacts with BRCA2 and inhibits the interaction between BRCA2 and RAD51 [170]. Interestingly, purified SYCP3 can bind RAD51 directly to inhibit its activity in D-loop formation but has no effect on DMC1 [218]. More studies are needed to understand how SYCP3 interacts with both BRCA2 and RAD51 and the impact on the BRCA2-RAD51 interactions in cancer cells and meiotic cells.

### 5.6. FANCD2, the Central Hub of the FA Pathway

FANCD2 forms a heterodimer with its paralogue, FANCI, and the complex is recruited to chromatin in response to DNA damage [171]. Then, FANCD2-FANCI is monoubiquitinated by the FA core complex in an ATR-dependent manner [171], an essential step required to recruit downstream FA/HR proteins during ICL repair including BRCA2 [172]. Both monoubiquitinated and non-ubiquitinated FANCD2 directly interacts with BRCA2 in human cells [42,173,174], and the interaction site was mapped to residues 2350–2545 in the helical domain of BRCA2 by yeast two-hybrid assays (Figure 2) [42].

FA patient-derived cell lines with FANCD2-deficiency display significantly lower IR-induced RAD51 foci formation with similar levels as those in FA patient-derived cell lines with BRCA2-deficiency. The recruitment defects of RAD51 and BRCA2 to the damage sites in these FANCD2-deficient cells were restored by expressing wild-type FANCD2 but not a mutant form of FANCD2 that cannot be ubiquitylated [174]. FANCD2 is required to protect replication forks from MRE11-dependent fork degradation, acting epistatically with RAD51 and BRCA2 [175]. FANCD2 also plays a role in pathway choice for replication restart, in a BRCA2-independent manner, to maintain fork stability [176,177,178]. Indeed, BRCA2-deficient tumor cells rely on FANCD2 for fork protection and cell survival [176,177]. Synthetic lethal interactions were reported between BRCA2 and FANCD2 [177], as well as between BRCA2 and POLθ [130,219]. FANCD2 promotes the recruitment of POLθ to the damaged sites and MMEJ repair for cell survival [177]. FANCD2 is also required in Pol kappa-mediated fork restart, as part of the activated FA complex [178]. Recently, a study showed that non-ubiquitylated FANCD2 functions together with BRCA2 and FANCJ to promote replication fork restart, independent of the FA core complex [179]. In summary, FANCD2 is a critical player in DNA repair and an essential influencer on the pathway choice at the replication fork.

### 5.7. Cyclin-Dependent Kinases

The functions of BRCA2 are regulated throughout the cell cycle via CDK-dependent phosphorylation [43]. Phosphorylation of BRCA2 at residue S3291 in the C-terminal RAD51 binding domain (TR2) increases through G2/M and is reduced in G1 [43]. This phosphorylation event signals mitotic entry by disrupting the BRCA2-RAD51 interaction and promoting RAD51 filament disassembly [43,120]. The phosphorylation mutant of BRCA2 S3291A, similar to the phosphomimic mutant S3291E, retains normal function in HR [43,79,120], but fails to protect stalled replication fork against nuclease degradation due to unstable RAD51 filament [79]. These results are consistent with the previous findings that TR2 is important to stabilize RAD51-ssDNA filaments [28,220]. Upon DNA damage induction, the cyclin A-CDK2-dependent phosphorylation of BRCA2 at S3291 is dramatically downregulated [43]. Cyclin D1 interacts directly with the C-terminus of both RAD51 and BRCA2, blocking access of cyclin A to BRCA2 and ensuing phosphorylation [221,222]. Cyclin D1 is overexpressed in several human cancers, and loss of cyclin D1 reduces damage-induced RAD51 focus formation, impairs HR repair, and sensitizes cancer cells to chemotherapeutic agents [222]. In response to replication stress, ATR activates the tumor suppressor RASSF1A, and RASSF1A activates the Hippo pathway kinase LATS1 to bind CDK2, inhibiting S3291 phosphorylation of BRCA2 and promoting RAD51 nucleoprotein filament formation for HR repair [223]. At stalled replication forks, BRCA2 needs to be phosphorylated at residue T77 by CDK to enable its binding to PLK1, serving as a platform to facilitate PLK1 to phosphorylate RAD51 at S14 and accumulate RAD51 [44]. In summary, BRCA2 activity in HR repair and RFP is carefully modulated through CDKs and likely additional kinases. We refer the reader to more comprehensive recent discussions of the posttranslational modifications of BRCA2 [224,225].

### 5.8. USP21

In addition to phosphorylation, BRCA2 is ubiquitylated in cells and subjected to proteasome-mediated degradation [185]. The first deubiquitinating enzyme (DUB) was identified as a BRCA2 binding partner is ubiquitin-specific protease 11, USP11. Although when overexpressed USP11 can deubiquitinate BRCA2 in vivo, it does not antagonize the MMC-induced ubiquitination and degradation of BRCA2 [185]. A second ubiquitin-specific protease, USP21, was found to deubiquitinate BRCA2 and stabilize its protein level upon genotoxic stress. The USP21 binding domain was mapped to the OB folds in the C-terminus of BRCA2 (Figure 2). As expected, the protein level of a C-terminal truncation mutant of BRCA2 lacking the OB folds was not altered by USP21 depletion. USP21-mediated BRCA2 stabilization promotes RAD51 loading at DSB sites. Cells overexpressing USP21 increased HR efficiency, and cells defective of USP21 showed a significant decrease in HR [33]. Overall, USP21 acts as a positive regulator of BRCA2 in HR, and it is directly involved in BRCA2 stabilization against proteasome-mediated degradation.

## 6. Pathogenic BRCA2 Mutations and BRCA2 Pseudo-Revertants

Loss-of-function mutations in the BRCA1/2 genes are associated with increased genome instability and predisposition to different but not all forms of cancers. Defects in BRCA2 binding partners and other HR genes result in tumors with similar HR deficiency, termed as “BRCAness”. These tumors initially respond well to DNA damage-based therapies such as ionizing radiation, cisplatin, and PARPi. PARPi exploits the synthetic lethality with BRCA-deficiency (or BRCAness) leading to selective killing of tumor cells [226,227]. This section discusses pathogenic BRCA2 mutants and BRCA2 pseudo-revertants as genetic tools to understand the essential domains of BRCA2 in maintaining genome stability, which ultimately will facilitate future clinical diagnosis and patient treatment.

To date, over 20,000 variants of BRCA2 have been documented in the BRCA Exchange [228], and over 13,000 variants have been reported in the ClinVar database [47]. Almost 3000 variants of BRCA2 in the BRCA Exchange and about 1/3 of the variants in the ClinVar are considered to be pathogenic. Most pathogenic BRCA2 mutations are small insertions or deletions that result in a frameshift, generating a premature stop codon and producing a truncated protein similar to a nonsense mutation. The majority (86%) of the pathogenic variants are frameshift and nonsense mutation, while the rest are splicing and missense variants. Analysis of the mutation distribution among breast and ovarian cancer families identified several regions along the BRCA2 gene with high risks. The ovarian cancer cluster regions (OCCR) largely overlap with the BRC repeats and the helical domain of BRCA2, whereas the breast cancer cluster regions (BCCR) cover both PALB2/EMSY-binding domains and DNA-binding domains [45,46]. Most of the pathogenic variants are located within these clusters, with the exception that some variants which are located in the C-terminal RAD51-binding domain (Figure 2). The mechanistic basis of why different regions of BRCA2 are hotspots for distinct cancers remains unclear.

PARPi and platinum-based chemotherapy have been used to effectively treat BRCA2 mutant tumors. However, accumulating evidence indicates that tumors develop resistant mechanisms during treatment, and a clinically relevant mechanism is the acquisition of secondary mutations of BRCA2 to regain some degree of BRCA2 activity. These BRCA2 pseudo-revertants can arise from base substitutions and insertions or deletions with various sizes, resulting in a restoration of the BRCA2 open reading frame or removal of the initial deleterious mutations [229,230]. The pancreatic cancer cell line CAPAN-1 features the most prevalent BRCA2 mutation *6174delT* lacking the C-terminal 1360 amino acids, including the last BRC repeat, the entire DBD, and the two NLSs [231]. Two of the PARP-inhibitor-resistant (PIR) clones identified from the CAPAN1 line, PIR1 and PIR2, have only recovered the extreme C-terminus containing the TR2 and NLS domains, compared to the original BRCA2 truncation mutant CAPAN1 *6174delT* [229]. Lacking the entire DBD and BRC6-8, PIR1 and PIR2 are still partly functional, as evidenced by restoration of some damage-induced RAD51 foci formation, reduced chromosomal aberrations, resistance to DNA damage agents, and loss of sensitivity towards PARPi [229]. It is likely that the PIR1 and PIR2 gained partial functions due to their intact protein interaction with PALB2, since the abolishment of PALB2 interaction completely inactivates a BRCA2 construct lacking the entire DBD in HR repair based on a DR-GFP assay [49]. It is unclear if additional genetic changes drive the phenotypes in these CAPAN1 pseudorevertants. The generation of pseudo-reversion mutations in BRCA2-deficient tumors reflects tumor evolution under selection. There are at least three contributing factors: increased mutation rate due to exposure to genotoxic agents, the lack of error-free DNA repair, and a selective advantage for BRCA1/2-restored cells during PARPi or platinum treatment [232]. Overall, the restored DNA repair function makes it challenging to target BRCA2 revertant tumors to overcome therapeutic resistance.

## 7. Conclusions

Mounting evidence has gradually unveiled the role of BRCA2 in DNA repair and replication fork support, as well as how its binding partners might modulate the functions of BRCA2 in mitosis and meiosis. Genome-wide sequencing has assembled a mutation signature for tumors lacking functional BRCA2 or HR repair pathway and provided a selection matrix to target these BRCAness tumors with precision therapy such as PARPi. However, the biochemical and structural mechanisms of how BRCA2 engages DNA, RAD51, and other binding partners remain largely unknown, hindering our understanding of how BRCA2 participates in HR and fork protection in diverse cellular contexts. It remains a puzzle that how multiple interacting partners interact with BRCA2 temporally and spatially to partition into different cellular events and respond to various cellular signals. More mechanistic studies are needed to understand these fundamental properties to provide the underpinning for future therapeutic approaches targeting BRCA2, its partners, and their synthetic lethal dependencies.

## Figures and Tables

**Figure 1 genes-12-01229-f001:**
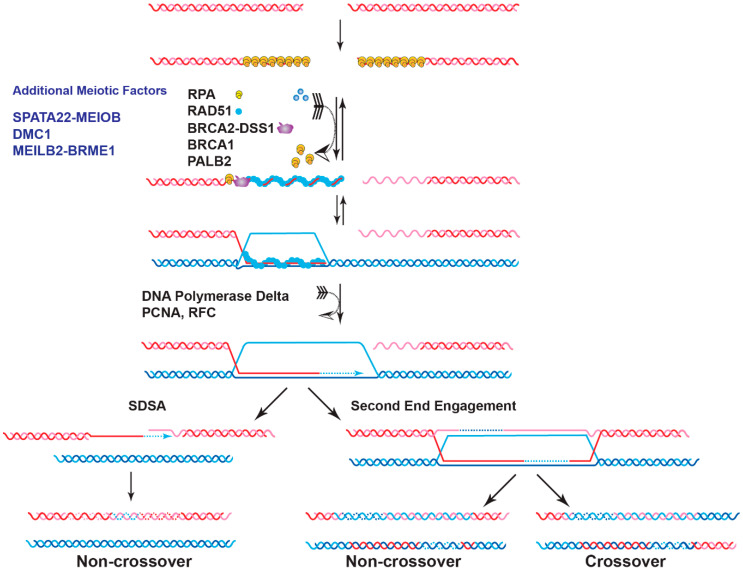
The mediator role of BRCA2 in homologous recombination. HR is a high-fidelity pathway to repair spontaneous DSBs in somatic cells and programmed DSBs in meiotic cells. DSB ends are first processed by nucleases to generate 3′-tailed ssDNAs, which are immediately covered by RPA. PALB2 interacts with BRCA1 and then recruits BRCA2 to the damaged sites. In meiosis, 3′-tailed ssDNA is bound by RPA and the meiosis-specific ssDNA binding complex SPATA22-MEIOB. BRCA2 is also recruited by meiosis-specific complex MEILB2/HSF2BP-BRME1. The key step of HR is the formation of filaments by RecA-type proteins on ssDNA (RAD51 in somatic cells, RAD51, and the meiosis-specific DMC1 during meiosis) with the simultaneous displacement of RPA and/or SPATA22-MEIOB. The nucleation of these filaments on RPA-coated ssDNA is entirely dependent on BRCA2. Assembled RAD51/DMC1 filaments catalyze homology search and DNA strand invasion of the dsDNA (sister chromatid in somatic cells, homologous chromosome in meiotic cells) to form displacement loops (D-loops). After new DNA synthesis and engaging the second end of the DSB, the joint molecule will be resolved or dissolved to produce crossover and non-crossover products (**Right**). In the Synthesis-Dependent Strand Annealing (SDSA) pathway, the extended D-loop will be disrupted to produce only non-crossover products (**Left**). In somatic cells, non-crossovers are preferred to avoid loss-of-heterozygosity, while in meiotic cells, one crossover per chromosome or chromosome arm is preferred to ensure accurate segregation during the first meiotic division and increase genetic diversity (see [15] for detailed discussions of the HR pathway). Meiosis-specific factors are labeled in blue, and only factors discussed in the text are annotated.

**Figure 2 genes-12-01229-f002:**
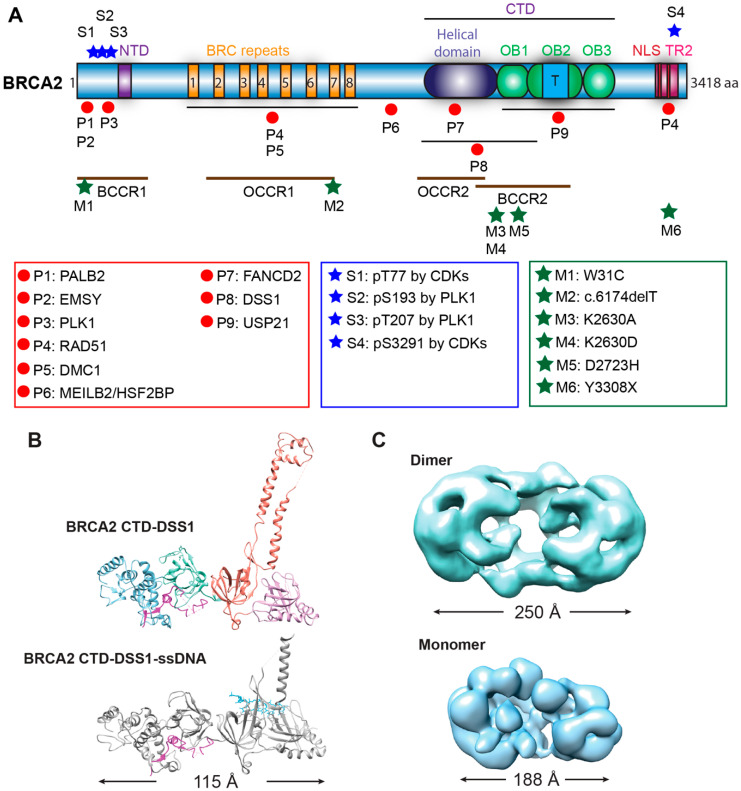
Structure of BRCA2. (**A**) Domain structures with mapped sites for selected binding partners and phosphorylation. BRCA2 has two DNA-binding domains (N-terminal DNA-binding domain or NTD, C-terminal DNA-binding domain or CTD), eight BRC repeats interacting with RAD51 and DMC1, a C-terminal RAD51 interaction domain (TR2), and two nuclear localization signals (NLS). Mapped interaction sites with selected protein partners are indicated with red circles and listed in the red box [10,11,12,33,34,35,36,37,38,39,40,41,42]. Selected phosphorylation sites with blue stars and sites are listed in the blue box [35,36,43,44]. The proposed breast and ovarian cancer cluster regions (BCCR and OCCR) are labeled with brown lines [45,46]. Several common pathogenic mutations (ClinVar database) and loss-of-function mutations are marked by green stars and listed in the green box [34,47,48,49]. Abbreviation: aa, amino acid; OB fold, Oligonucleotide/Oligosaccharide-Binding fold; T, tower domain (three-helical bundle); P, binding protein; S, phosphorylation site; M, pathogenic mutation site; BCCR, breast cancer cluster region; OCCR, ovarian cancer cluster region. (**B**) Crystal structures of BRCA2 CTD in a complex with DSS1 alone (top, 1MIU) or with both DSS1 and ssDNA (bottom, 1MJE) [12]. (**C**) Low-resolution electron microscopic (EM) structures of the dimeric full-length BRCA2 (top, EMD-2779) [30] and the monomeric BRCA2-DSS1-ssDNA complex (bottom, EMD-21998) [23].

**Figure 3 genes-12-01229-f003:**
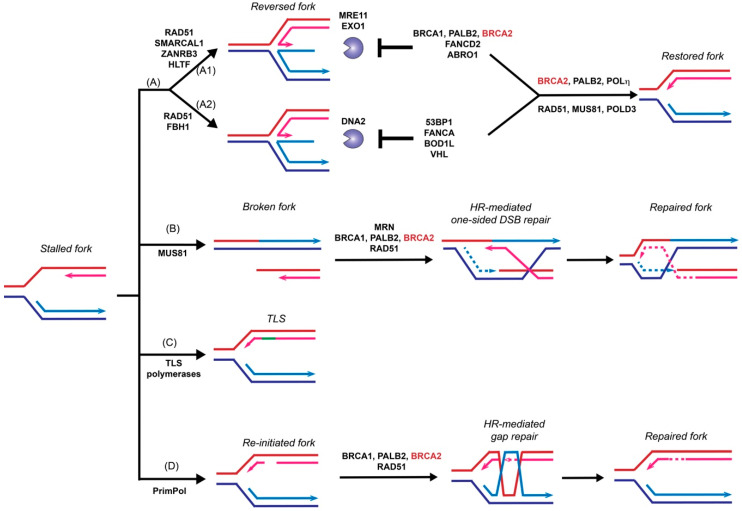
Roles of BRCA2 in replication fork restart. Under replication stress, stalled or collapsed forks can be restarted by several pathways, including fork reversal (**A**), homologous recombination to repair one-sided double-stranded breaks formed after fork collapse (**B**), fork repriming by translesion DNA synthesis bypassing the lesion (**C**), or by PrimPol-mediated replisome re-initiation downstream of the lesion followed by template switching to repair the associated gap (**D**). BRCA2 plays both homologous recombination (HR)-dependent and HR-independent roles to support fork restart and maintain fork stability. In fork reversal (**A**), stalled forks are reversed by RAD51 in two pathways involving DNA translocases (**A1**: SMARCAL1, ZANRB3, and HLTF; **A2**: FBH1)) to form chicken foot structures, which are susceptible to nuclease degradation by MRE11, EXO1, and DNA2. BRCA2 (in association with BRCA1 and PALB2), FANCD2, and ABRO1 protect reversed forks against MRE11- and EXO1-mediated resection (**A1**), while 53BP1, FANCA, BOD1L, and VHL prevent DNA2-mediated degradation (**A2**). Whether the two pathways generate alternate DNA structures is not known. Reversed forks are restored primarily by the RAD51, MUS81, POLD3 pathway and secondarily by the BRCA2, PALB2, and POLη pathway. The repair of one-sided DSBs (**B**) by HR is presumed to occur in a fashion highly analogous to the repair of two-sided DSBs (see Figure 1) including the critical mediator function of BRCA2 in RAD51 filament formation. BRCA2 appears to play no role in translesion DNA synthesis (**C**). Finally, BRCA2 plays a crucial role to facilitate RAD51-mediated HR to repair DNA gaps left behind re-initiated forks (**D**) by template switching. Arrowheads represent the 3′-OH end.

**Table 1 genes-12-01229-t001:** BRCA2 binding partners and their cellular functions.

Protein	Binding Location	BRCA2-Related Function	Cellular Function	References
RAD51	BRC repeats and TR2 region	- BRCA2-mediated filament formation to search homology sequences in mitosis and stimulate DMC1 filament formation in meiosis. - BRCA2-mediated filament formation to inhibit nucleolytic degradation of stalled forks.	HR RFP	[4,5,18,19,20,21,22,27,28,37,38,79,101,147]
DMC1	BRC repeats and TR2 region	- BRCA2-mediated filament formation to search for homology sequences in meiosis.	Meiotic HR	[18,39,81,83,84,85,96]
PALB2	Residue 21–39	- BRCA2 localizer to DNA damaged sites	HR RFP	[11,34,119,122,141,142,148]
EMSY	Residue 23–44	- Unknown	HR? Transcription factor?	[10,149,150,151]
DSS1	Helical + OB1 + OB2	- Control BRCA2 self-association, protein stability, nuclear localization, RPA removal	DNA repair Protein degradation RNA metabolism	[12,19,23,48,152,153,154,155,156,157,158,159,160,161,162,163,164,165,166]
MEILB2	Residues 2117–2339	- BRCA2 localizer to DNA damaged sites in meiosis. - BRCA2 negative regulator in ICL repair.	Meiotic HR	[40,41,93,167]
SYCP3	Unknown	- A component of synaptonemal complex. - BRCA2 negative regulator in somatic HR.	Meiotic HR	[168,169,170]
FANCD2	Residues 2350–2545	- BRCA2 recruiter in ICL repair.	ICL repair RFP	[42,171,172,173,174,175,176,177,178,179]
FANCG	Unknown	- Mediating the formation of BRCA2-FANCD2-FANCG-XRCC3 complex in ICL repair.	ICL repair	[173]
XPG	Unknown	- Mediating presynaptic filament formation.	HR NER	[180]
BCCIP	Residues 2973–3001	- Stimulating BRCA2 and RAD51 foci formation.	HR Chromosome segregation	[181,182,183,184]
Pol η	Residues 1338–1781	- BRCA2 and PALB2-dependent fork recovery.	Fork stability TLS	[122]
CDK2	Residue 3291	- Control of BRCA2 RFP function via phosphorylation of S3291	Cell-cycle control	[43,79]
PLK1	Residues S193, T207	- Phosphorylate BRCA2 T207 site to form BRCA2-PLK1-BUBR1-PP2A complex, which stabilize kinetochore-microtubule attachments. - Phosphorylate BRCA2 S193 site to form BRCA2-Nonmuscle myosin IIC (NM-IIC) complex at the midbody, which is required for precise midbody abscission.	Chromosome segregation Cytokinesis	[35,36,44,143,146]
USP21	OB folds	- Stabilizing BRCA2 protein.	Deubiquitylation	[33,185]
BRAF35	BRC5	- Regulate cytokinesis, transcription, cell-cycle progression.	Cytokinesis Transcriptional repression Cell-cycle progression	[139,140,186]
p/CAF	Residues 290–453	- Spindle assembly checkpoint	Chromosome segregation	[144]
BUBR1	Residues 3149–3418	- Spindle assembly checkpoint	Chromosome segregation	[144]
Filamin A	Residues 2516–3030	- BRCA2-mediated localization at the midbody to stimulate the CEP55 signaling pathway.	Cytokinesis	[145,146]
CEP55	Residues 421–982	- Stimulating the formation of abscission complex at the midbody.	Cytokinesis	[145,146]
Nonmuscle myosin IIC (NM-IIC)	Unknown	- BRCA2-mediated formation of II-C ring at the midbody, which is required for precise midbody abscission.	Cytokinesis	[146]
p53	BRC repeats and OB2 + OB3 domains	- Suppression of p53 transcription activity. - HR repression.	Transcription control	[187]

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
