# Peer review of "Guardians of the Genome: BRCA2 and Its Partners"

_genes, 2021, doi:10.3390/genes12081229_

Round 1

Reviewer 1 Report

see attached.

Reviewer 2 Report

The manuscript genes-1296084 under the title “Guardians of the genome: BRCA2 and its partners” represents an interesting review article that discusses the fundamental biochemical and structural properties of BRCA2 and describes how these properties contribute to mechanisms of DNA repair and replication fork support in mitotic and meiotic living cells. Important BRCA2 binding partners and functional regulators and their role in BRCA2-mediated DNA repair mechanisms are also highlighted as well as the important pathogenic mutations and pseudo-revertants of the BRCA2 gene.

 The entire manuscript is very well written in all its parts, sufficiently and qualitatively illustrated, and properly referenced.

I believe that the data presented in this review will be quite interesting and valuable for the readers of the Genes journal.

Regarding the request for additional comments.

The list of abbreviations used in the text should be provided
Furthermore, all abbreviations mentioned in the figures should be described in the corresponding figure legends
Also, I have noticed that numbers of references follow the text but some of the references mentioned in the figure legends may be enumerated accordingly since they appear before the corresponding ones in the manuscript text. Although I can understand the logic of the authors regarding the used reference numbers in the text and figure legends.
